# Development of Androgen-Antagonistic Coumarinamides with a Unique Aromatic Folded Pharmacophore

**DOI:** 10.3390/ijms21155584

**Published:** 2020-08-04

**Authors:** Hitomi Koga, Mai Negishi, Marie Kinoshita, Shinya Fujii, Shuichi Mori, Mari Ishigami-Yuasa, Emiko Kawachi, Hiroyuki Kagechika, Aya Tanatani

**Affiliations:** 1Department of Chemistry, Faculty of Science, Ochanomizu University, 2-1-1 Otsuka, Bunkyo-ku, Tokyo 112-8610, Japan; Kogahitomi.ocha@gmail.com (H.K.); mai.negishi0903@gmail.com (M.N.); aimer61@icloud.com (M.K.); 2Institute of Biomaterials and Bioengineering, Tokyo Medical and Dental University (TMDU), 2-3-10 Kanda-Surugadai, Chiyoda-ku, Tokyo 101-0062, Japan; fujiis.chem@tmd.ac.jp (S.F.); s-mori.chem@tmd.ac.jp (S.M.); myuasa.chem@tmd.ac.jp (M.I.-Y.); e.k.sakuratomomochan@gmail.com (E.K.); 3Institute of Human Life Innovation, Ochanomizu University, 2-1-1 Otsuka, Bunkyo-ku, Tokyo 112-8610, Japan

**Keywords:** prostate cancer, cell proliferation, coumarin, cis-amide

## Abstract

First-generation nonsteroidal androgen receptor (AR) antagonists, such as flutamide (**2a**) and bicalutamide (**3**), are effective for most prostate cancer patients, but resistance often appears after several years due to the mutation of AR. Second-generation AR antagonists are effective against some of these castration-resistant prostate cancers, but their structural variety is still limited. In this study, we designed and synthesized 4-methyl-7-(*N*-alkyl-arylcarboxamido)coumarins as AR antagonist candidates and evaluated their growth-inhibitory activity toward androgen-dependent SC-3 cells. Coumarinamides with a secondary amide bond did not show inhibitory activity, but their *N*-methylated derivatives exhibited AR-antagonistic activity. Especially, **19b** and **31b** were more potent than the lead compound **7b**, which was comparable to hydroxyflutamide (**2b**). Conformational analysis showed that the inactive coumarinamides with a secondary amide bond have an extended structure with a *trans*-amide bond, while the active *N*-methylated coumarinamides have a folded structure with a *cis*-amide bond, in which the two aromatic rings are placed face-to-face. Docking study suggested that this folded structure is important for binding to AR. Selected coumarinamide derivatives showed AR-antagonistic activity toward LNCaP cells with T877A AR, and they had weak progesterone receptor (PR)-antagonistic activity. The folded coumarinamide structure appears to be a unique pharmacophore, different from those of conventional AR antagonists.

## 1. Introduction

Androgen receptor (AR) is a ligand-dependent transcription factor belonging to the nuclear receptor superfamily [1,2], and its endogenous ligands, so-called androgens, are testosterone (**1a**) (see Appendix A) and dihydrotestosterone (DHT, **1b**) (Figure 1). These androgens have various roles in the differentiation, growth, and maintenance of the male reproductive organs [3,4]. Based on its physiological and pharmacological functions, AR is recognized as an important target for drug discovery. For example, since androgen plays an important role in the progression of prostate cancer, AR antagonists have been used clinically for the treatment of prostate cancer [5,6,7]. However, AR antagonists bearing a steroid core structure have undesirable side effects due to their cross-activity with other steroid hormone receptors. Consequently, nonsteroidal AR antagonists have been developed, and some of them, such as flutamide (**2a**) [8] and bicalutamide (BIC, **3**, Figure 1) [9,10], are in clinical use to treat prostate cancer. These first-generation nonsteroidal AR antagonists are effective for most prostate cancer patients, but castration-resistant prostate cancer (CRPC) often develops after a few years, due to the mutation of AR [11,12,13]. For example, T877A is the most common mutation of AR, and hydroxyflutamide (**2b**), an active metabolite of flutamide (**2a**), acts as an agonist toward T877A AR [14]. Bicalutamide (**3**) acts as antagonist toward T877A AR, but as an agonist toward W741C AR [15].

Second-generation AR antagonists effective against CRPC have been developed, such as enzalutamide (**4**) [16,17], apalutamide [18,19], and darolutamide [20]. Structurally, these AR antagonists have the same pharmacophore as the lead compounds, flutamide (**2a**) and bicalutamide (**3**), and they contain an anilide structure with electron-withdrawing functional groups on the phenyl ring (Figure 1). Recently, non-anilide type AR antagonists have also been reported, but variation in the chemotype is still limited [21].

We have reported 6-arylcoumarin derivatives such as **5** and **6** as novel nonsteroidal progesterone receptor (PR) antagonists (Figure 2) [22,23]. In these compounds, the coumarin ring provides an alternative hydrophobic core structure to the sterane ring of steroidal derivatives or to the bicyclic anilide scaffold of known nonsteroidal PR ligands (i.e., benzoxadinone, and benzimidazolone derivatives bearing a nitrogen atom in the bicyclic heterocyclic skeleton). Among them, compounds **5**, **6b**, and **6c** showed moderate AR-antagonistic activity in the androgen-dependent SC-3 cell growth assay [24]. These results suggested that the coumarin ring could mimic the anilide scaffold of AR antagonists. In addition, we reported 4-benzyl-1-(2*H*)-phthalazinone derivatives as AR antagonists with a novel core structure [25]. With this background, we decided to search for compounds with AR-antagonistic activity among our coumarin compound library, and we found that compound **7b** bearing a 7-(*N*-methyl-benzoylamino) moiety showed AR-antagonistic activity with rather low PR-antagonistic activity (Figure 2). Compound **7b** has a unique core structure that is different from those of conventional AR antagonists. In this study, we examined the structure–activity relationship of coumarinamide derivative **7b** as an AR antagonist and identified a unique pharmacophore, consisting of an aromatic folded core structure that can interact with amino acid residues in the ligand-binding pocket of AR.

## 2. Results and Discussion

Various 4-methyl-7-(*N*-methyl-arylcarboxamido)coumarins with different substituents on the phenyl ring or amide nitrogen atom were synthesized from 7-amino-4-methylcoumarin by *N*-acylation, followed by *N*-alkylation (Scheme 1); their structures are shown in Table 1, Table 2 and Table 3. In order to clarify the electronic and steric properties of the amide bond, the biological activities of the derivatives bearing a secondary amide bond (series **a**) were also examined.

AR-agonistic and antagonistic activities of the synthesized coumarinamide derivatives were evaluated in terms of growth-inhibitory activity toward SC-3 cells bearing wild-type (wt) AR [24]. These cells show androgen-dependent cell proliferation, which is one of the typical and reliable assays for AR modulators [26]. None of the coumarinamide derivatives examined affected the proliferation of SC-3 alone at a concentration below 10^−5^ M (data not shown), which means they do not act as AR agonists. The antagonistic activity of the test compounds was examined in terms of their effect on 1 nM DHT-dependent proliferation of SC-3 cells. The IC_50_ values of test compounds are shown in Table 1, and the dose-dependency of the selected active compounds are shown in Figure 3. Compound **7b**, the lead compound in this study, inhibited the DHT-induced proliferation of SC-3 cells with the IC_50_ value of 1.34 µM, and its activity is nearly one-fifth that of hydroxyflutamide (**2b**; IC_50_ = 0.29 μM), the positive control.

First, we examined the substituent effect on the phenyl ring of **7b**. Among the synthesized compounds, most of the coumarinamides with a secondary amide bond (series **a**) did not affect SC-3 cell growth, except for compounds **16a** and **19a**, which showed weak inhibitory activity only at 10 μM. On the other hand, some of the *N*-methylated compounds (series **b**) dose-dependently inhibited the DHT-induced proliferation of SC-3 cells. The introduction of a substituent at the para position of the phenyl group was not effective, and only compound **11b** bearing a small fluorine atom exhibited inhibitory activity, being more potent than the lead compound without a substituent on the phenyl ring, and comparable in potency to hydroxyflutamide (**2b**). The introduction of a substituent at the ortho position was also ineffective, while compounds bearing a *meta*-substituent showed potent inhibitory activity. The introduction of a *meta*-cyano group, which is an electron-withdrawing group common to the structures of bicalutamide (**3**) and enzalutamide (**4**), decreased the activity compared to the lead compound **7b**. The introduction of a halogen atom or halogen-containing substituent at the meta position increased the activity, and compounds **16b**, **17b**, and **18b** showed more potent activity than **7b**. Among the compounds with a *meta*-substituent, compound **19b** with the *m*-methyl group was most active in SC-3 assay, showing an IC_50_ value of 0.49 μM. Introduction of a second *meta*-substituent was not effective, and compounds **24b** and **25b** with *m,m’*-dichloro or *m,m’*-dimethyl groups, respectively, are slightly less active than the corresponding monosubstituted compounds **18b** and **19b**.

Secondly, the effect of *N*-substituents was examined for the selected compounds **18b** and **19b** (Table 2). In both cases, as the *N*-substituent became larger, the inhibitory activity decreased. Thus, the **c** and **d** compounds with a *N*-ethyl or *N*-*n*-propyl group showed moderate inhibitory activity, while the **e** and **f** compounds with the *N*-substituent containing an aromatic ring were less active.

Thirdly, we replaced the phenyl group of compound **7b** with other ring structures (Table 3). Compound **26b** with a saturated cyclic alkyl group instead of the phenyl ring was inactive, which was probably due to the bulkiness of the nonplanar cyclohexyl ring. Interestingly, compound **27b** with a 1-naphthyl group is more potent than the lead compound **7b**, while compound **28b** with a 2-naphthyl group, an isomer of **27**, exhibited weaker inhibitory activity. Thus, the orientation of the naphthyl group is important for the activity. The derivatives of **7b** with *para*-substituents were almost inactive (Table 1), although the para position of compound **28b** could be regarded as substituted by a benzo group. Replacement of the phenyl ring of **7b** with the basic pyridine ring diminished the activity, while replacement with 5-membered heterocyclic aromatics such as furyl and thiophenyl groups increased the activity. Compound **31b** had potent inhibitory activity with the IC_50_ value of 0.50 µM, which was comparable to that of hydroxyflutamide (**2b**).

Next, we examined the activity of the selected compounds **18b** and **19b** toward LNCaP cells bearing mutated T877A AR (Figure 4) [27,28]. Bicalutamide (**3**) inhibited the proliferation of LNCaP cells, whereas hydroxyflutamide (**2b**) did not. Compounds **18b** and **19b** inhibited the DHT-dependent cell growth of LNCaP cells, and their inhibitory activities were higher than that of bicalutamide (**3**). Thus, coumarinamide derivatives can act as AR antagonists toward mutated T877A AR.

As discussed above, the coumarinamide derivatives (series **a**) with a secondary amide bond (R = H) showed little or no activity, while their *N*-methylated derivatives (series **b**) exhibited potent AR-antagonistic activity. We have previously reported on the unique conformational properties of aromatic amide compounds [28,29]. Thus, secondary amides such as benzanilide existed in trans form, while their *N*-methylated amides exist in cis form in the crystal and predominantly in cis form in solution. The cis conformational preference is a general property of *N*-methylated amides, including those bearing *N*-heterocyclic aromatics such as pyridine, pyrrole, and imidazole. Therefore, we assumed that the difference in AR-antagonistic activity between secondary and *N*-methylated compounds would result from the difference in their conformational properties. Then, we examined the crystal structures of compounds **7a** and **7b** (Figure 5). As expected, compound **7a** exists in the extended structure with a *trans*-amide form, in which the phenyl and coumarin rings are located on opposite sides of the molecule. On the other hand, compound **7b** exists in a folded structure with *cis*-amide, in which the two aromatic rings take a face-to-face position. Comparison of the ^1^H NMR chemical shifts of **7a** and **7b** indicated that these compounds exist in solution in conformations similar to those found in the crystal. Thus, compounds **7b** shows aromatic proton signals at a higher field than compound **7a** in the ^1^H NMR spectra, due to the anisotropic effects between two aromatic rings. A similar tendency was observed with the other coumarinamide derivatives. These results indicate that the folded structure with a *cis*-amide bond is important for AR-antagonistic activity.

In order to analyze the binding features of coumarinamide derivatives with AR, we conducted a docking study of compounds **7a** and **7b** with the AR ligand-binding domain (LBD) by using the AutoDock program. For the docking, we used the crystal structure of the AR LBD in the complex with DHT (**1b**) (PDB ID: 2AMA) and the crystal structures of **7a** and **7b**, as shown in Figure 6. Notably, **7b** with the folded structure can dock with the AR LBD in a similar position to that of DHT (**1b**). In this binding structure, the carbonyl group of the *cis*-amide bond of compound **7b** lies in the same direction as that of DHT (**1b**), and the phenyl group of **7b** is placed in the hydrophobic region. On the other hand, compound **7a** with an extended structure can bind to the ligand-binding pocket, but there is no strong interaction with the surrounding amino acid residues. The binding energy of **7a** with AR LBD is –7.55 kcal/mol, which is lower in energy than that of **7b** (−9.04 kcal/mol). Thus, the docking study also supports the significance of the folded coumarin amide structure for AR-antagonistic activity.

Finally, we examined the activity of selected coumarinamide derivatives toward progesterone receptor (PR), since some coumarin derivatives act as PR antagonists. The PR-antagonistic activities were evaluated by means of alkaline phosphatase (AP) assay using the T-47D human breast carcinoma cell line (Figure 7) [30]. All the selected coumarinamide derivatives examined showed PR-antagonistic activity only at 10 µM, and they did not affect the progesterone (1 nM)-induced AP activity at concentrations below 1 µM. In particular, the PR-antagonistic activity of compound **31b** was very weak, and the compound only slightly inhibited AP activity, even at 10 µM. Thus, the coumarinamide derivatives show selectivity as AR antagonists over PR.

## 3. Materials and Methods

### 3.1. Chemistry

#### 3.1.1. General

All reagents were purchased from Sigma-Aldrich Chemical Co. (Tokyo, Japan), Tokyo Kasei Kogyo Co. (Tokyo, Japan), Wako Pure Chemical Industries (Tokyo, Japan), or Kanto Kagaku Co., Inc. (Tokyo, Japan). Silica gel for column chromatography was purchased from Kanto Kagaku Co., Inc (Tokyo, Japan). ^1^H and ^13^C NMR spectra were recorded on a JEOL ECA 600 (Tokyo, Japan), or Bruker 600 spectrometer (Tokyo, Japan). Mass spectral data were obtained on a Bruker Daltonics microTOF-2focus (Tokyo, Japan), Thermo Scientific Q-Exactive or Waters Q-TOF Premier in the positive ion detection mode (Tokyo, Japan).

#### 3.1.2. Synthesis of **23a** (General Procedure for Coumarinamides with Secondary Amide Bond)

*ο*-Anisoyl chloride (120 mg, 0.70 mmol) was added to a solution of 7-amino-4-methylcoumarin (101 mg, 0.58 mmol) in pyridine (3 mL), and the mixture was stirred at 50 °C for 2 h. The mixture was poured into 2 M HCl (30 mL), and the precipitates were collected, washed with water, and a mixture of chloroform and methanol to give **23a** (171 mg, 96%).

#### 3.1.3. Synthesis of **23b** (General Procedure for N-Alkylated Coumarinamides)

Sodium hydride (24 mg, 0.60 mmol) was washed with *n*-hexane. A solution of **23a** (61 mg, 0.20 mmol) in DMF (1 mL) was added to a suspension of sodium hydride in DMF (2 mL). After 20 min, iodomethane (83 mg, 0.58 mmol) was added to the mixture. After 5 h, the reaction mixture was poured into water and extracted with ethyl acetate. The organic layer was washed with brine, dried over magnesium sulfate, and concentrated. The residue was purified by silica gel column chromatography (ethyl acetate:hexane = 1:1) to give **23b** (21.0 g, 32%).

### 3.2. SC-3 Growth Inhibition Assay

SC-3 cell growth inhibition assay was performed as the previous report [24,26]. SC-3 cells were seeded in 96-well plates at a concentration of 2000 cells per well in 100 μL of MEMα medium supplemented with 2% FBS (fetal bovine serum), and the plates were incubated for 24 h at 37 °C under 5% CO_2_. Then, 10 μL of medium was removed from each well and replaced with 10 μL of drug solution supplemented with serial dilutions of test compound or DMSO as a dilution control in the presence of 1 nM DHT, and the plates were incubated for 3 days. The cell number was determined using a Cell Counting Kit-8 (Dojindo, Kumamoto, Japan) according to the manufacturer’s instruction. This parameter is proportional to the number of living cells in the culture. IC_50_ (Table 1, Table 2 and Table 3) is the concentration of test compound that reduces DHT-induced cell growth to 50% of the control. All experiments were performed in triplicate or more.

### 3.3. LNCaP Cell Proliferation Assay

LNCaP cell growth inhibition assay was performed as the previous report [27,28,31]. Briefly, LNCaP cells were seeded in 96-well plates at a concentration of 2000 cells per well in 100 μL of RPMI-1640 medium supplemented with 2% charcoal-stripped FBS, and the plates were incubated for 24 h at 37 °C under 5% CO_2_. Then, 10 μL of medium was removed from each well and replaced with 10 μL of drug solution supplemented with serial dilutions of test compound or DMSO as a dilution control in the presence or absence of 10 nM DHT. Cells were incubated for 6 days, and half of the medium was removed and replaced once after 3 days with medium containing test compound or DMSO as a dilution control. At the end of the incubation, proliferation was evaluated using a Cell Counting Kit-8 (Dojindo) according to the manufacturer’s instruction. This parameter is proportional to the number of living cells in the culture. All experiments were performed in triplicate or more.

### 3.4. Alkaline Phosphatase Assay Using T-47D Cells

Alkaline phosphatase assay was performed based on the previous report [22,32]. Briefly, T-47D cells were plated in 96-well plates at 10,000 cell per well and incubated at 37 °C under 5% CO_2_ for 24 h. Then, 10 μL of medium was removed from each well and replaced with 10 μL of drug solution supplemented with serial dilutions of test compound or DMSO as a dilution control in the presence or absence of 1 nM progesterone, and incubation was continued for 24 h. Then, the medium was aspirated, and the cells were fixed with 100 μL of 1.8% formalin in PBS. The fixed cells were washed with PBS, and 75 μL of assay buffer (1 mg/mL *p*-nitrophenol phosphate in diethanolamine water solution, pH 9.0, 2 mM MgCl_2_) was added. The mixture was incubated at room temperature with shielding from light for 2 h, and then the reaction was terminated by the addition of 100 μL of NaOH. The absorbance at 405 nm was measured with a microplate reader.

## 4. Conclusions

We synthesized a series of 4-methyl-7-(*N*-alkyl-arylcarboxamido)coumarins as AR antagonist candidates, using compound **7b** as a lead compound. The AR-antagonistic activity of the synthesized coumarinamide derivatives was examined in terms of growth-inhibitory activity toward SC-3 cells. Coumarinamides with a secondary amide bond were inactive, but their *N*-alkylated derivatives exhibited AR-antagonistic activity. Among them, compounds **19b** and **31b** showed more potent AR-antagonistic activity than the lead compound **7b**, and they were comparable in potency to hydroxyflutamide (**2b**). Conformational analysis and docking study revealed that the folded structure of coumarinamides associated with the *cis*-amide bond is important for binding to AR. The selected coumarinamide derivatives showed AR-antagonistic activity toward LNCaP cells with T877A AR and had weak PR-antagonistic activity. The detailed action mechanism of novel coumarinamides as AR modulators including AR-binding ability and transcriptional regulation should be elucidated in the future works, as the folded coumarinamide is a different pharmacophore from those of conventional AR antagonists, and these compounds may be promising candidates for the treatment of prostate cancer.

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
