# Peer review of "Development of Androgen-Antagonistic Coumarinamides with a Unique Aromatic Folded Pharmacophore"

_ijms, 2020, doi:10.3390/ijms21155584_

Round 1
Reviewer 1 Report
Koga et al designed and synthetized new AR antagonists candidates and evaluated their growth-inhibitory activity on wild-type (SC-3) and T877A (LNCAP) AR expressing prostate cells.
I do not have major commentss concerning the work. My recommendation is minor revision.
I have only one minor remark. In tables 1, 2 and 3, the IC50 antiandrogen values o are indicated in M. They should be replaced by µM.
Author Response
We revised the manuscript according to the reviewers’ comments as follows. The revised parts are shown in red color in the text of the manuscript file.
Comments from Reviewer #1
- Reviewer’s comment: Koga et al designed and synthetized new AR antagonists candidates and evaluated their growthinhibitory activity on wild-type (SC-3) and T877A (LNCAP) AR expressing prostate cells. I do not have major comments concerning the work. My recommendation is minor revision. I have only one minor remark. In tables 1, 2 and 3, the IC50 antiandrogen values o are indicated in M. They should be replaced by μM.
Our response: Thank you for your comments. This is our simple mistake, and revised the unit of concentration of IC50 in Tables 1 – 3.
- Reviewer’s comment:
(x) English language and style are fine/minor spell check required
Our response: We carefully checked English language and style of our manuscript.
Reviewer 2 Report
The manuscript is well written and the study design is linear.
The results may have a relevant clinical impact.
Author Response
We revised the manuscript according to the reviewers’ comments as follows.
Comments from Reviewer #2
- Reviewer’s comment: The manuscript is well written and the study design is linear. The results may have a relevant clinical impact.
(x) English language and style are fine/minor spell check required
Our response: Thank you for your comments. We carefully checked English language and style of our manuscript.
Reviewer 3 Report
Authors prepared large series of new coumarinamides and successfully tested them as androgen antagonists. Great benefit of this work is strong multidisciplinary approach, which can enable to preparation of novel nonclassical pharmacophore and describe the mechanism of their effect. About my opinion these compounds represent perspective agents for cancer treatment. Also, their synthesis biological studies and calculation method is described well. And thereby I can recommend publication of presented manuscript in this journal.
Author Response
We revised the manuscript according to the reviewers’ comments as follows.
Comments from Reviewer #3
- Reviewer’s comment: Authors prepared large series of new coumarinamides and successfully tested them as androgen antagonists. Great benefit of this work is strong multidisciplinary approach, which can enable to preparation of novel nonclassical pharmacophore and describe the mechanism of their effect. About my opinion these compounds represent perspective agents for cancer treatment. Also, their synthesis biological studies and calculation method is described well. And thereby I can recommend publication of presented manuscript in this journal.
(x) English language and style are fine/minor spell check required
Our response: Thank you for your comments. We carefully checked English language and style of our manuscript.